# Design and Analysis of Modified U-Shaped Four Element MIMO Antenna for Dual-Band 5G Millimeter Wave Applications

**DOI:** 10.3390/mi14081545

**Published:** 2023-07-31

**Authors:** Chandrasekhar Rao Jetti, Tathababu Addepalli, Sreenivasa Rao Devireddy, Gayatri Konni Tanimki, Ahmed Jamal Abdullah Al-Gburi, Zahriladha Zakaria, Pamarthi Sunitha

**Affiliations:** 1Department of ECE, Bapatla Engineering College (A), Bapatla 522102, Andhra Pradesh, India; 2Department of ECE, Aditya Engineering College, Surampalem 533437, Andhra Pradesh, India; 3Department of ECE, Koneru Lakshmaiah Education Foundation, Vaddeswaram 522302, Andhra Pradesh, India; 4Center for Telecommunication Research & Innovation (CeTRI), Fakulti Teknologi Kejuruteraan Elektrikal dan Elektronik (FTKEE), Universiti Teknikal Malaysia Melaka (UTeM), Ayer Keroh 75450, Malaysia; 5Centre of Telecommunication Research & Innovation (CeTRI), Fakulti Kejuruteraan Elektronik dan Kejuruteraan Komputer (FKEKK), Universiti Teknikal Malaysia Melaka (UTeM), Durian Tungal 76100, Malaysia

**Keywords:** 5G mmWave, impedance matching, isolation, mutual coupling and multiple input multiple output (MIMO)

## Abstract

A novel compact-slotted four element multiple input multiple output (MIMO) planar monopole antenna is proposed for 5G mmWave N257/N258 and N262 band applications. The antenna, with dimensions of 12 mm × 11.6 mm × 0.508 mm ($1.036\lambda_{o}\;\times$
$1.001\lambda_{o}\times 0.043\lambda_{o}$ where $\lambda_{o}$ is computed at lowest cutoff frequency), is fabricated on a Rogers RT/duroid 5880 (tm) substrate with a relative permittivity of 2.2 and a dielectric loss tangent of 0.0009. The suggested antenna consists of four U-shaped radiating elements (patches) on top of the dielectric material and a slotted ground on the bottom. The radiating elements are fed by a 50-ohm microstrip line feed. To improve the impedance performance of the MIMO antenna, a rectangular strip of 1.3 mm × 0.2 mm and a couple of rectangular slots are added to each radiating element. The first operating band at 27.1 GHz, ranging from 25.9 GHz to 27.8 GHz, is achieved by using slotted U-shaped radiating elements. The second operating band at 48.7 GHz, ranging from 47.1 GHz to 49.9 GHz, is obtained by etching hexagonal slots on the ground. The antenna design achieves an isolation of >27 dB through the orthogonal positioning of radiating elements and slots on the ground. The designed antenna operates at 27 GHz (N257/N258) and 48.7 GHz (N262) bands, exhibiting stable radiation patterns, a peak gain of >5.95 dBi, radiation efficiency of >90%, an envelope correlation coefficient of <10^−6^, a total active reflection coefficient of ≤−10 dB, channel capacity losses of <0.03 bits/s/Hz, and a mean effective gain of ≤−3 dB. The simulated and measured results of the antenna show good agreement, making it well-suited for 5G mmWave communication applications.

## 1. Introduction

Wireless communication networks are experiencing widespread usage, leading to an increased demand for significantly higher data rates, minimal latency, and efficient spectrum utilization in portable device applications. The advent of 5G (fifth generation) mobile communication technology offers several advantages over the current 4G (fourth generation) system, including higher transmission rates, improved spectrum efficiency, and reduced latency. Notably, 5G networks boast data rates that are 10 times higher than those of 4G networks, along with enhanced device connectivity and stability. The FCC has divided the 5G frequency spectrum into three bands: a low-band, sub-6 GHz band, and mmWave band. The mmWave frequency range, extending beyond 24 GHz, provides ample spectrum availability, enabling exceptional capacity, high throughput, and remarkably low latency. Consequently, various 5G mmWave antenna models have been proposed [1,2,3,4,5]. For 5G applications, multiple input multiple output (MIMO) technology is a potential solution. In a MIMO system, multiple antennae are employed at both ends of the communication channel. MIMO technology has the capability to enhance data transmission and spectral efficiency without requiring additional bandwidth or transmit power. However, due to the close spacing of antenna elements, mutual coupling can occur in MIMO systems. Consequently, improving isolation or reducing mutual coupling between adjacent antenna presents a challenge [6,7,8,9]. Extensive research has already been conducted to address isolation improvement and mutual coupling reduction in MIMO antenna systems [10,11,12,13,14,15,16,17,18,19,20,21,22,23,24,25,26,27,28,29,30].

For 2-element MIMO antenna systems, several approaches have been developed to enhance isolation [10,11,12,13,14,15,16,17,18,19,20]. These approaches include the use of rectangular ground slots [10], inverted antenna elements with high spacing [11], a rectangular micro-strip stub with a defected ground plane [12], narrow and T-shaped slots on the ground plane [13], DGS with a T-shaped stub on the ground [14], orthogonal arrangement of antenna elements [15], diversity in antenna element placement [16], antenna with multiple layers having different dielectric constants [17], inverted arrangement of radiators [18,19], and double-side EBG (DS-EBG) structure [20]. Furthermore, various methods for improving isolation in four element MIMO systems have been established [21,22,23,24,25,26,27,28,29,30,31,32,33]. These methods include the integration of a low-pass filter in the antenna structure [21], a tapered slot on the ground plane [22], a new rounded patch electromagnetic band gap (EBG) cell [23], two repetitions of regular split-ring slots on a partial ground [24], and integrating circular and semi-circular shaped gaps as well as a cross-shaped ground changed with a prolonged circular-shaped flaw [25], a defective ground with a circular, rectangular, and zigzag-shaped slotted arrangement [26], a slotted zigzag decoupling assembly imprinted from edge to edge on the top side of the patch [27], utilizing two successive iterations of ground structures with defects [28], a basic geometric parasitic element loaded in between the MIMO elements [29], the use of a partial ground plane [30], incorporating ladder resonator [31], use of orthogonal mode pairs [32] and orthogonal placement of radiating elements [33]. While these aforementioned antennae offer high isolation, they tend to be large and have complicated designs. As a result, 5G mmWave communication requires a simple and compact MIMO antenna with enhanced isolation.

This research introduces a novel compact dual-band U-shaped MIMO design for 5G mmWave systems. The suggested design comprises four U-shaped radiating elements on top of a substrate with a slotted ground on the bottom. Each radiating element of the MIMO antenna is equipped with a rectangular strip and a pair of rectangular slots to enhance its impedance performance. The first operating band at 27.1 GHz is achieved using slotted U-shaped radiating elements, while the second operating band at 48.7 GHz is created by incorporating hexagonal slots into the ground plane. Additional isolation is provided by the orthogonal positioning of the antenna elements and narrow rectangular slots in the ground. The developed antenna exhibits consistent radiation patterns, high peak gain, increased radiation efficiency, negligible envelope correlation coefficient (ECC), acceptable total active reflection coefficient (TARC), minimal channel capacity losses (CCL), and low mean effective gain (MEG) at 27.1 GHz (N257/N258) and 48.7 GHz (N262) bands. The following subsections provide a detailed explanation of the proposed MIMO antenna.

## 2. Antenna Design and Analysis

Figure 1 shows the proposed dual-band MIMO antenna for 5G mmWave N257/N258 and N262 band applications. The top view, rare view, isometric view, and isometric 3D view with optimal dimensional parameters are depicted in Figure 1a–d. Ansoft HFSS was used to perform the proposed antenna design, dimension optimization, and simulations. The antenna is fabricated on the Rogers RT/duroid 5880(tm) dielectric with 12 × 11.6 mm^2^, which has a thickness of 0.508 mm and a relative permittivity of 2.2. The antenna is fed via a 50-ohm microstrip line of sizes 1.035 × 0.3 mm^2^. The suggested antenna is made up of four U-shaped radiating elements (R1 to R4) on top of a dielectric material and slotted ground at the bottom. Each radiating element is formed by combining the sub-elements of dimensions 2.25 × 0.8 mm^2^, 1.565 × 0.8 mm^2^, and 4.59 × 1.145 mm^2^. The outer dimensions of the sub-elements of a patch are denoted by 3.395 mm, 4.59 mm, and 2.71 mm. Also, to enhance the MIMO antenna’s impedance performance (denoted by Sii, i = 1 to 4), two rectangular slots (1.465 × 0.325 mm^2^ and 0.78 × 0.325 mm^2^) and a rectangular strip of 1.3 × 0.2 mm^2^ are added to each radiating patch. Using rectangular slots and a strip, first resonance was induced in the 27.1 GHz band, which covers from 25.9 GHz to 27.8 GHz as depicted in Figure 2a. The first operating frequency spectrum includes N257 and N258 mmWave frequency bands in the 5G spectrum. The elements R1 and R3 are 0.205 mm from the substrate’s edge, whereas the elements R2 and R4 are 0.105 mm from the substrate’s edge. A ground plane with slots in the shapes of hexagons and rectangles makes up the substrate’s bottom layer as represented in Figure 1b. The size of each hexagonal slot is 2.8 × 2.425 mm^2^. Etching hexagonal slots on the ground plane results in the second operational band at 48.7 GHz from (47.1–49.9) GHz as shown in Figure 2a. The second band covers the N262 mmWave frequency band in the 5G spectrum. The hexagonal slots beneath the R1 and R3 elements are 1.287 mm and 1 mm from the dielectric material’s borders, respectively. Similarly, the hexagonal slots under the R2 and R4 elements are 1.196 mm and 1.11 mm from the dielectric material’s corners, respectively. 

The orthogonal arrangement of radiating elements and the right selection of distance between the radiating elements improve the isolation. The distance between the elements is 2.775 mm and 2.475 mm, as given in Figure 1a. In addition, From Figure 1b, four tiny rectangular slots with the size 4.7 × 0.3 mm^2^ are carved out of the ground plane to produce excellent isolation or minimal mutual coupling between the antenna elements. The rectangular slots are cut 0.1 mm from a corner of the substrate and slots are situated 5.85 mm and 5.65 mm from the substrate’s edges. Figure 2a shows the return loss characteristics of the proposed design. According to Figure 2b, the suggested MIMO antenna achieves isolation (denoted by S21/S31/S41) of >27dB (or mutual coupling <−27 dB) using orthogonally positioned radiating elements and rectangular slots on the ground plane.

The surface current distribution parameter is one of the crucial visual parameters for MIMO antenna systems. It demonstrates how the current of one element affects another in the MIMO design. Figure 3 describes the surface currents of the proposed antenna at 27.1 and 48.7 GHz resonating frequencies. It can be observed that most of the surface current is clustered surrounding the narrow rectangular slots, with a small amount of current pouring to the remaining ports; hence, isolation was enhanced between the elements.

### 2.1. Single Element Design and Evolution

The single-element design and evolution process is described in Figure 4. The proposed antenna evolution stages S11 responses are plotted in Figure 5. Antenna #A is the basic rectangular antenna of size 4.59 × 3.395 mm^2^ (C × E mm^2^). Antenna #A operates at 27 GHz and 51.4 GHz which does not include the desired bands as from Figure 5. By cutting a rectangular slot of dimensions 2.99 × 2.25 mm^2^ and 0.685 × 0.8 mm^2^ from Antenna #A, Antenna #B is formed. Antenna #B only works at 27 GHz (N257/N258 band). To attain the second band at 48.7GHz (N262 band) in addition to the 27 GHz band, a hexagonal slot was cut from the ground plan. Moreover, to achieve impedance-matching properties at working bands, a couple of slots and a strip are used on the radiator. So, Antenna #C is adopted as the basic radiating element in the suggested MIMO antenna. The single antenna evolution stages’ current distribution on the surface is given in Figure 6.

### 2.2. Effect of Feed Width on Impedance Performance

In this subsection, the impact of feed width on antenna impedance performance is investigated. Figure 7 illustrates the S11 response of various feed widths such as 0.3, 0.5 and 0.7 mm. For the feed widths of 0.5 and 0.7 mm, the antenna resonates only at 27.1 GHz, as observed from Figure 7. The required dual bands at 27.1 and 48.7 GHz are realized only with a feed width of 0.3mm and hence is chosen in this work.

### 2.3. Effect of Ground Slot on Impedance Performance

The effect of the ground slot on impedance performance is also examined in this section. The S11 responses of the different ground slots like a hexagon, octagon, and decagon are plotted in Figure 8. The desired bands with a good impedance matching performance are achieved with hexagonal ground slots when compared with octagonal and decagonal slots. Therefore, a hexagonal ground slot is preferred in this design.

### 2.4. Effect of Slots on Isolation Performance

Two approaches are used in this work to provide excellent isolation between antenna ports which include an orthogonal arrangement of U-shaped radiating elements and the use of narrow rectangular ground slots. Here, the S-parameters S21, S31, and S41 are used to analyze the effect of narrow rectangular slots on the isolation performance. Figure 9 presents the isolation responses of the suggested antenna with and without slots. It is found that the antenna with slots offers very good isolation when compared with the antenna without slots.

## 3. Results and Discussions

### 3.1. S-Parameters

To evaluate the recommended MIMO antenna’s properties, a prototype was developed on a Rogers RT/duroid 5880 (tm) substrate and experimentally verified through Agilent N5224A VNA, as shown in Figure 10. An examination of measurement and simulation findings was carried out, considering several performance indicators such as impedance matching (Sii, i = 1 to 4) and isolation (Sij, i = j, where i ≠ j) characteristics. Figure 10a,b shows a comparison of simulation and measurement values from a constructed prototype. The proposed design operates at 27.1 GHz from (25.9–27.8) GHz and 48.7 GHz from (47.1–49.9) GHz frequencies with acceptable impedance matching and isolation of above 26dB. The results from the simulation and the measurements clearly show good agreement. The simulation and measurement S-parameter values agree well; however, there is a minor difference due to SMA connection losses, fabrication tolerances, dielectric, and conducting losses. Hence, the designed antenna is well-suitable for 5G mmWave N257/N258 and N262 band applications.

### 3.2. Radiation Performance

The radiation performance of an antenna is one of the most crucial factors as it indicates the direction, strength, and nature of the radiated fields. Figure 11a,b depicts the radiation characteristics (2D and 3D) of the proposed MIMO antenna at 27.1 GHz and 48.7 GHz when the antenna is loaded with a 50-ohm load from port-2 to port-4 and excited at port-1. The antenna demonstrates an omnidirectional H-pattern in both frequency bands, with slight deviations caused by the measurement arrangement. Additionally, the antenna achieves a maximum gain of 6.03 dBi at 27.1 GHz and 5.95 dBi at 48.7 GHz, as observed from the 3D polar plots. Figure 11c displays the fabricated prototype in an anechoic chamber for testing.

Figure 12 depicts the suggested design’s radiation efficiency and peak gain plots. The simulated efficiency on both working bands exceeds 90%. The measured peak gain corresponds well with the simulation. In the band of 27.1 GHz, the measured gain is approximately 5.9 dBi and the simulated gain is 6.03 dBi. And in the band of 48.7 GHz, the modelled gain is 5.95 dBi while the tested gain is about 5.5 dBi.

### 3.3. MIMO Performance

This section analyzes the diversity performance of the suggested MIMO antenna using several metrics of performance such as the envelope correlation coefficient (ECC), diversity gain (DG), total active reflection coefficient (TARC), channel capacity loss (CCL), and mean effective gain (MEG). The MIMO antenna diversity is analyzed using the ECC [34,35,36,37]. Equation (1) describes the ECC, which indicates the way the channels relate to one another for an N-element system. ECC of 0 denotes a lack of correlation and 1 denotes a high degree of correlation among the signals. The satisfactory value of ECC in real applications is <0.5. The simulated and validated ECC for designed four port dual-band MIMO antenna are shown in Figure 13a. The ECC value in the first and second bands is less than 10^−6^, indicating substantial agreement in the simulated and measured values.
(1)$${ECC}_{S\text{-}Parameters}=\frac{{\left|S_{ii}^{*}S_{ij}+S_{ji}^{*}S_{jj}\right|}^{2}}{\left(1-\left({\left|S_{ii}\right|}^{2}+{\left|S_{ij}\right|}^{2}\right)\right)\left(1-\left({\left|S_{jj}\right|}^{2}+{\left|S_{ij}\right|}^{2}\right)\right)}$$
(2)$$DG=10\sqrt{1-{\left|{ECC}_{S\text{-}Parameters}\right|}^{2}}$$
(3)$$TARC=\sqrt{\frac{\left|{\left(S_{11}+S_{12}e^{j\theta }\right)}^{2}\right|+\left|{\left(S_{21}+S_{22}e^{j\theta }\right)}^{2}\right|}{2}}$$
(4)$$C_{Loss}=-{log}_{2}det\left(\alpha^{R}\right)$$
where,
$$\alpha^{R}=\left[\begin{array}{cc} \alpha_{ii} & \alpha_{ij}\\ \alpha_{ji} & \alpha_{jj} \end{array}\right]\;$$
(5)$$\alpha_{ii}=1-\left({\left|S_{ii}\right|}^{2}+{\left|S_{ij}\right|}^{2}\right);\;\alpha_{ij}=-\left({S_{ii}}^{*}S_{ij}+{S_{ji}}^{*}S_{jj}\right);$$
(6)$$\alpha_{ji}=-\left({S_{jj}}^{*}S_{ji}+{S_{ij}}^{*}S_{ii}\right);\;\alpha_{ii}=1-\left({\left|S_{jj}\right|}^{2}+{\left|S_{ji}\right|}^{2}\right)$$
(7)$${MEG}_{i}={0.5\eta }_{i,rad}=0.5\left[1-\sum_{j=1}^{M}{\left|S_{ij}\right|}^{2}\right]$$

The diversity gain is an indicator of how well antenna diversity methods perform [34,35,36,37]. The suggested MIMO antenna diversity capability can be evaluated using Equation (2). The graphs of simulated and observed diversity gain with frequency are shown in Figure 13b. The proposed antenna yields a diversity gain of 10 dB across the functioning band and there is a good correlation between the two results.

Individual MIMO antenna S-parameters will not provide an exact analysis. The total active reflection coefficient (TARC) can be utilized to estimate the performance of MIMO systems. TARC takes into account self and mutual impedance variations [38]. The TARC for a MIMO system is determined using Equation (3). Simulated and measured TARC <−10 dB are provided throughout the whole band by the suggested MIMO design, as seen from Figure 13c.

The assessment of the channel capacity loss (CCL) is also necessary for the design of MIMO antennae. The formulas in Equations (4)–(6) can be used to determine this parameter as a function of the S-parameters [32]. The CCL simulation and measurement findings at the required frequency bands are shown in Figure 13d. The CCL is below 0.03 bit/s/Hz throughout the operational frequency bands. A diversity performance indicator known as the mean effective gain (MEG) indicates an antenna’s capacity to collect electromagnetic signals in a multipath environment. [39]. This parameter of the suggested antenna is computed with Equation (7). The calculated and recorded MEG plot is shown in Figure 13e. It can be noticed that both functional bands have a MEG of −3dB.

The recommended MIMO antenna performance has been compared to other antennae that have been previously described in Table 1. It is evident that most MIMO antenna configurations discussed in [10,11,12,13,14,15,16,17,18,19,20,21,22,23,24,25,26,27,28,29,30] are larger than the suggested antenna, thereby limiting their suitability for usage with modern compact devices. When compared to the existing antennae, it is evident that the recommended design outperforms them in terms of size, impedance bandwidth, isolation, peak gain, ECC, DG, TARC, CCL, and MEG. The designs in [27,28] offer high isolation and antennae in [21,27] provide more gain when compared with the proposed antenna. However, the offered isolation and gain are acceptable in practical applications. Therefore, the proposed antenna design demonstrates superior performance compared to previous designs, making it suitable for current and future communication systems.

## 4. Conclusions

For 5G mmWave N257/N258 and N262 band applications, a miniaturized dual-band slotted four element MIMO planar monopole antenna is proposed. The antenna design consists of four U-shaped radiating elements on top of the dielectric material and a slotted ground on the bottom. To enhance the impedance performance of the MIMO antenna, a rectangular strip and a pair of rectangular slots are added to each radiating element. The first operating band at 27.1 GHz, ranging from 25.9 GHz to 27.8 GHz, is achieved using slotted U-shaped radiating elements. The second operating band at 48.7 GHz, ranging from 47.1 GHz to 49.9 GHz, is obtained by etching hexagonal slots on the ground plane. The orthogonal positioning of the radiating elements and the presence of narrow rectangular slots on the ground provide isolation of more than 27 dB. The designed antenna operates at 27 GHz (N257/N258) and 48.7 GHz (N262) bands, exhibiting stable radiation patterns, high peak gain, increased radiation efficiency, very low envelope correlation coefficient (ECC), acceptable total active reflection coefficient (TARC), negligible channel capacity losses (CCL), and low mean effective gain (MEG). The proposed antenna is modeled, manufactured, and its results are measured. The simulation and test findings demonstrate good agreement, indicating that the proposed antenna is a suitable choice for 5G mmWave N257/N258 and N262 band applications.

## Figures and Tables

**Figure 1 micromachines-14-01545-f001:**
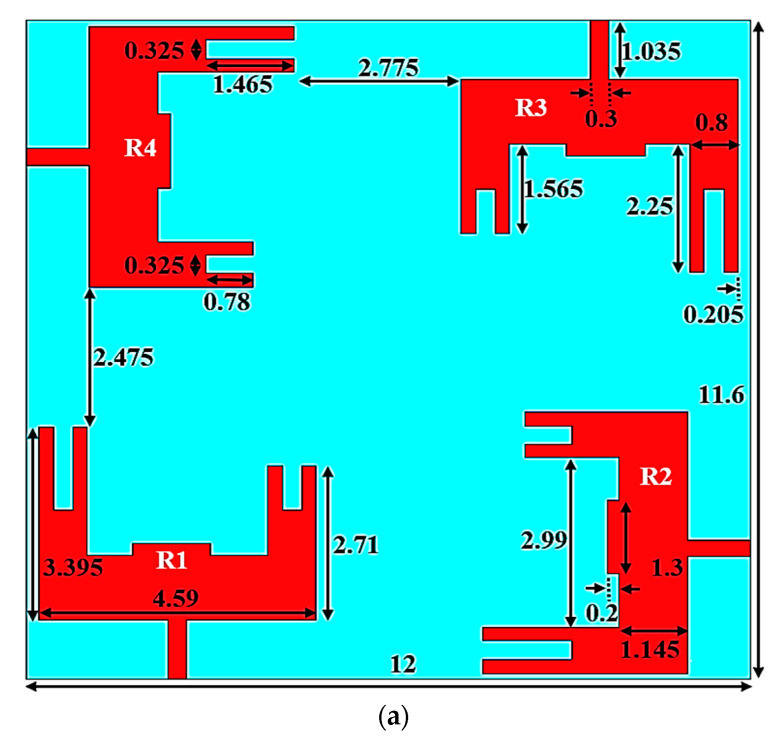
Proposed MIMO Antenna with Dimensional Parameters. (**a**) Top View, (**b**) Rare View, (**c**) Isometric View, and (**d**) Isometric 3D- View.

**Figure 2 micromachines-14-01545-f002:**
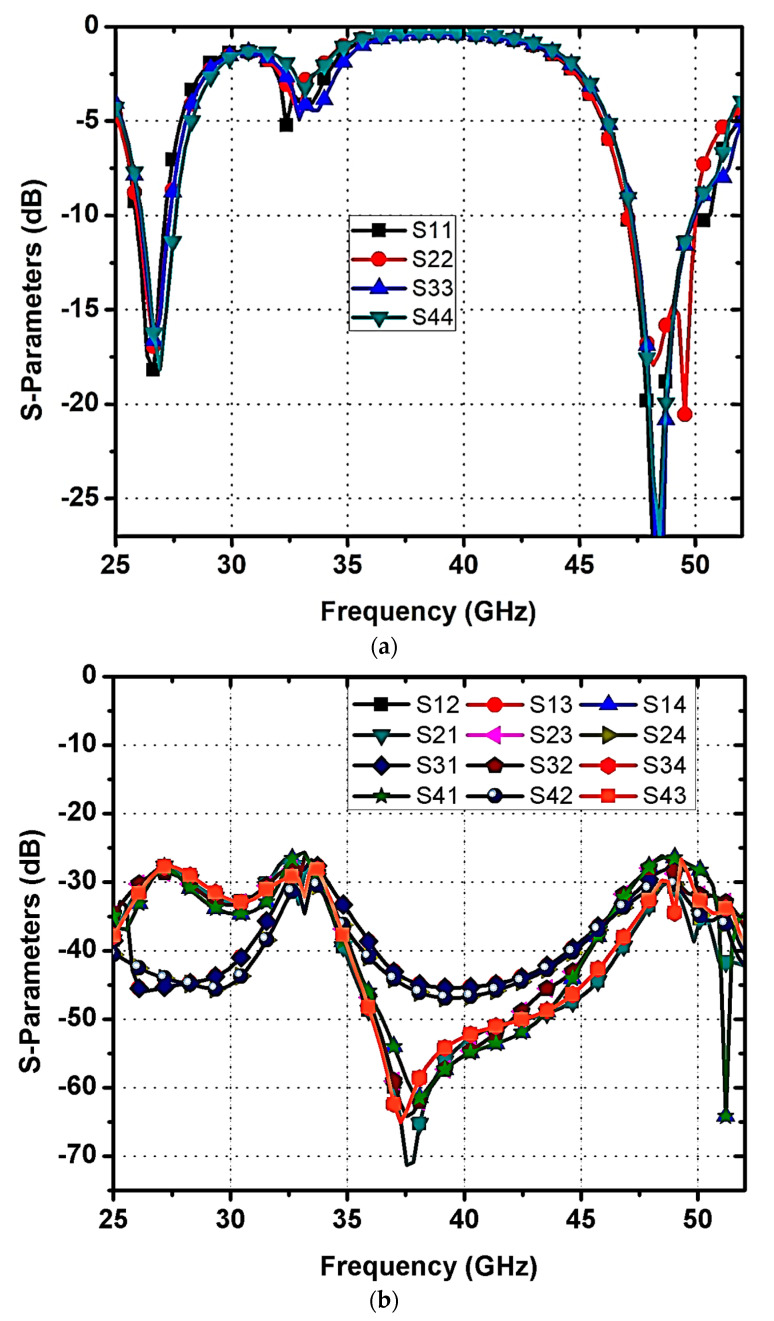
Proposed MIMO Antenna. (**a**) S11 and (**b**) S-Parameters (S11, S21, S31, and S41).

**Figure 3 micromachines-14-01545-f003:**
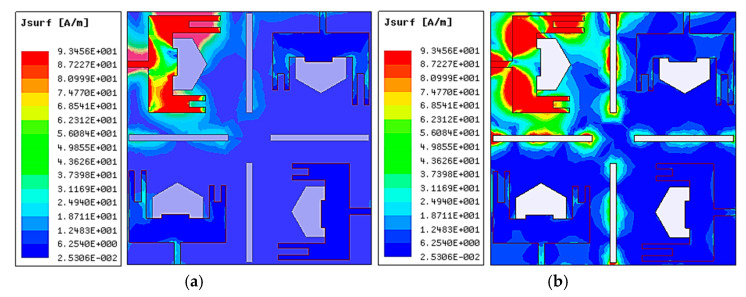
Proposed MIMO Antenna Surface Current Distribution. (**a**) At 27.1 GHz and (**b**) At 48.7 GHz.

**Figure 4 micromachines-14-01545-f004:**
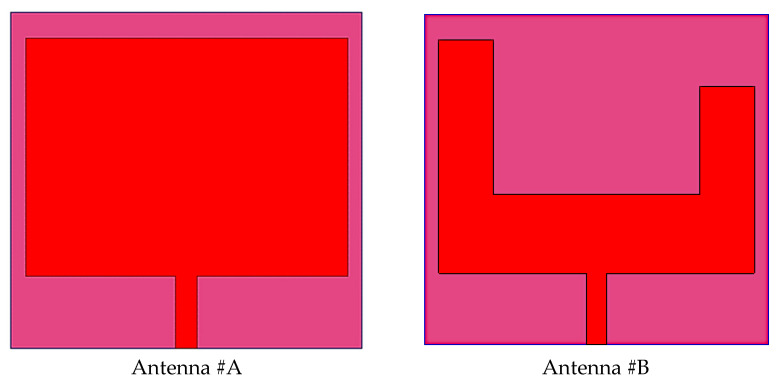
Evolution Stages of the Suggested MIMO Antenna.

**Figure 5 micromachines-14-01545-f005:**
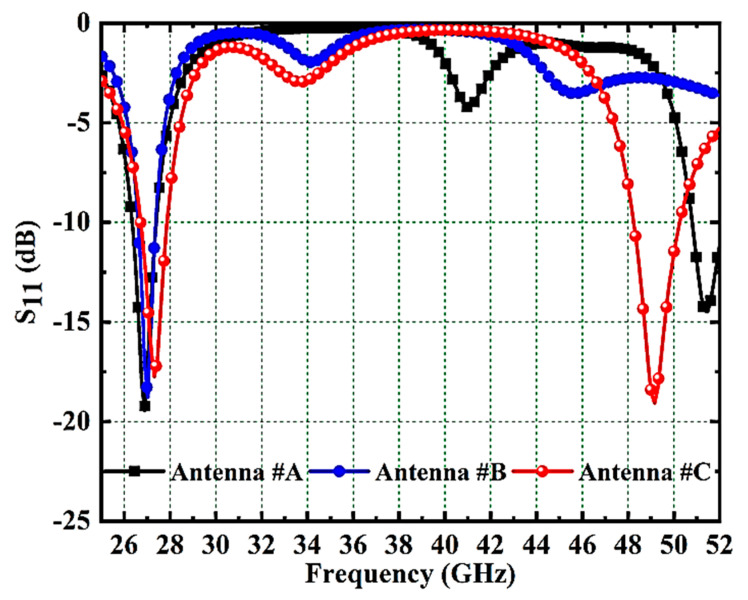
Proposed MIMO Antenna Evolution Stages’ S11 Responses.

**Figure 6 micromachines-14-01545-f006:**
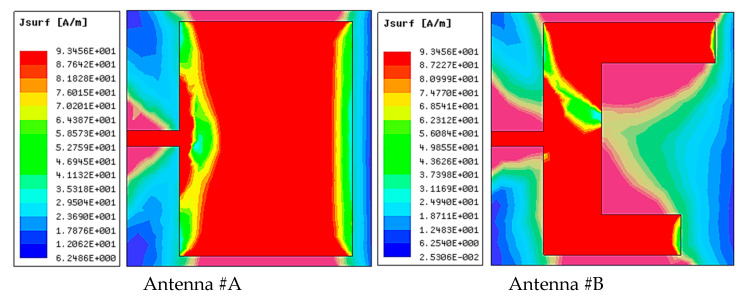
Proposed MIMO Antenna Evolution Stages’ Surface Current Distributions.

**Figure 7 micromachines-14-01545-f007:**
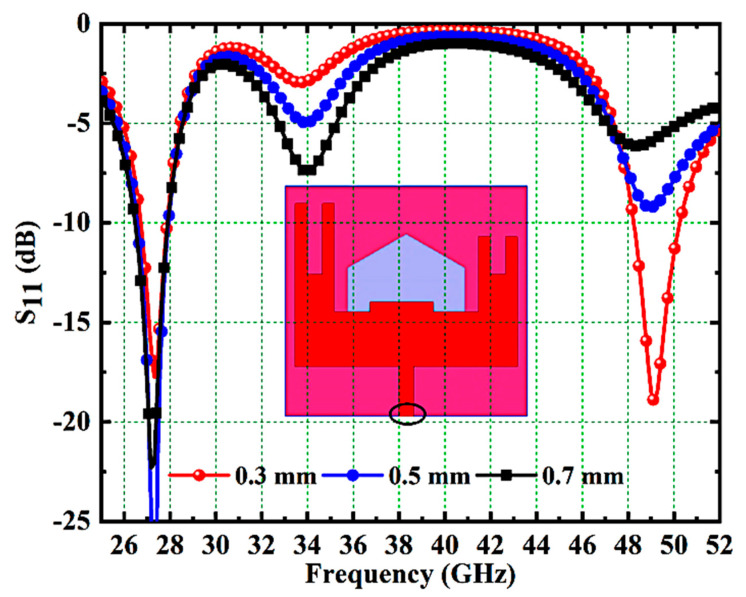
S11 Response of Various Feed Widths.

**Figure 8 micromachines-14-01545-f008:**
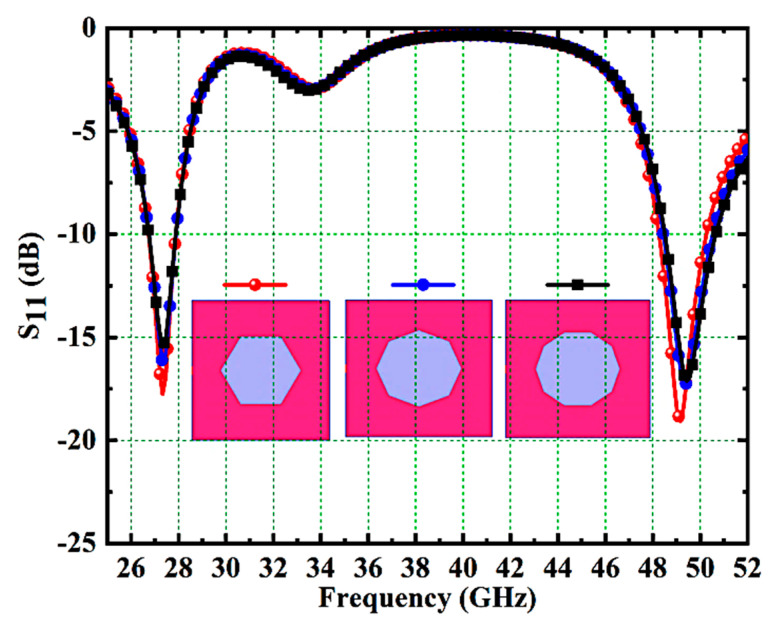
S11 Response of Various Ground Slots.

**Figure 9 micromachines-14-01545-f009:**
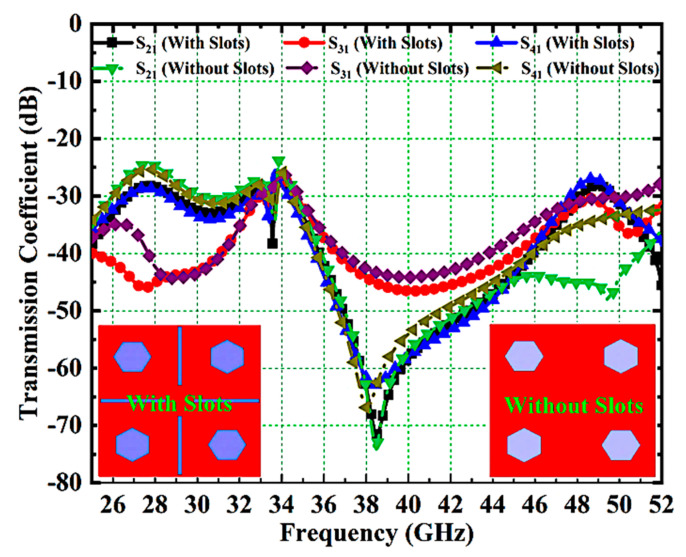
Isolation Response of Proposed MIMO Antenna With and Without Slots.

**Figure 10 micromachines-14-01545-f010:**
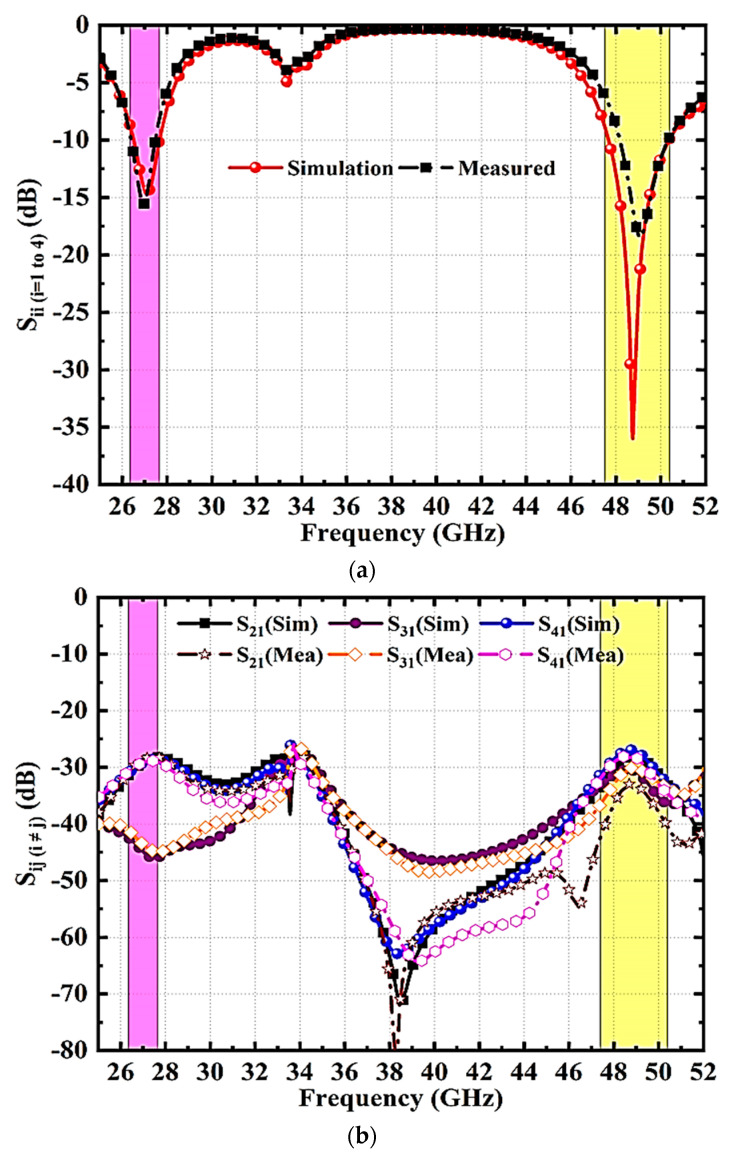
Proposed MIMO Antenna. (**a**) Sii Simulation and Measures Results Comparison, (**b**) Sij Simulation and Measures Results Comparison, (**c**) Prototype Top View, and (**d**) Prototype Rare View.

**Figure 11 micromachines-14-01545-f011:**
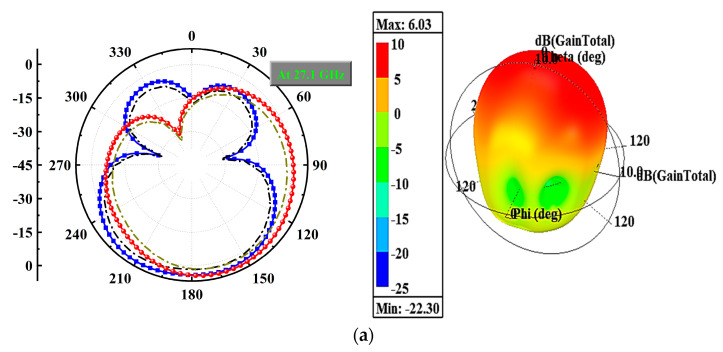
Proposed MIMO Antenna Radiation (2D and 3D) Characteristics. (**a**) 27.1 GHz, (**b**) 48.7 GHz, and (**c**) Fabricated Prototype in Anechoic Chamber for Testing.

**Figure 12 micromachines-14-01545-f012:**
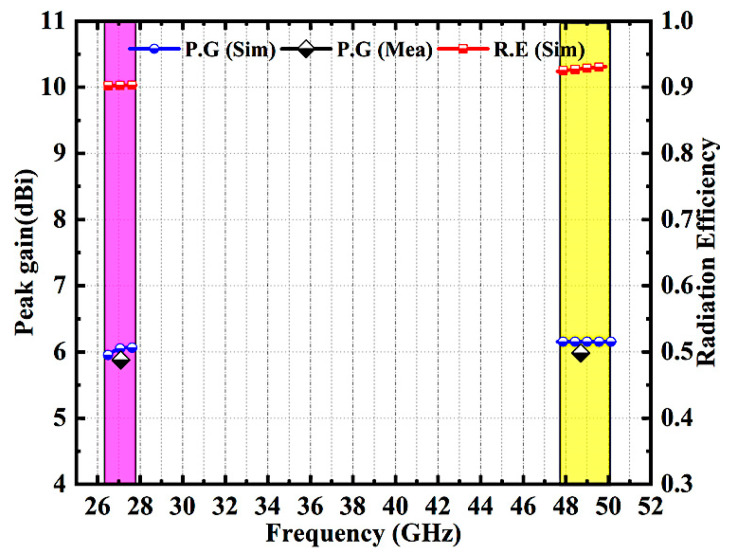
Proposed MIMO Antenna Radiation Efficiency and Peak Gain Characteristics.

**Figure 13 micromachines-14-01545-f013:**
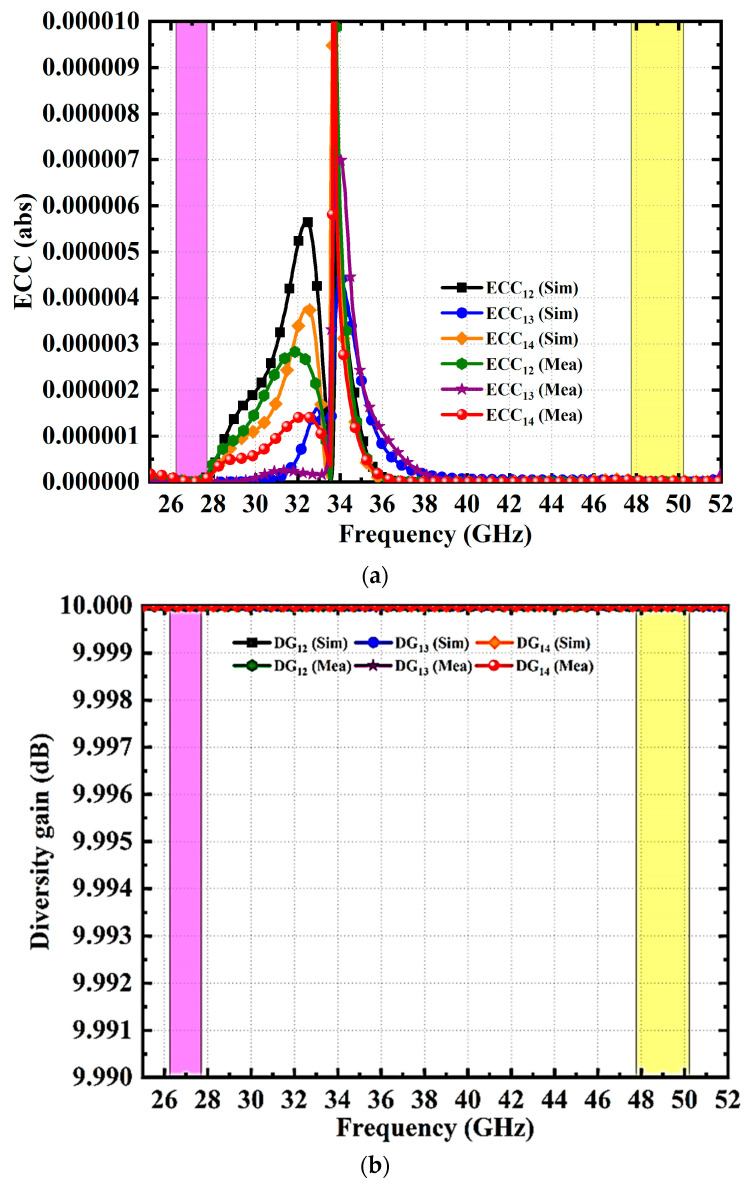
Proposed MIMO Antenna Diversity Performance (Simulation and Measured) Characteristics. (**a**) ECC, (**b**) DG, (**c**) TARC, (**d**) CCL, and (**e**) MEG.

**Table 1 micromachines-14-01545-t001:** Comparison of the reported and suggested MIMO antennae’s performance.

Ref.	No., of Elements	Size (in mm^3^)	Bands (GHz)	Isolation (dB)	Gain (dBi)	R.E (%)	ECC	DG (dB)	TARC (dB)	CCL (bits/s/Hz)	MEG (dB)
[10]	2	30 × 15 × 0.25	28	>35.8	5.42	-	<0.005	9.99	-	<0.1	<−3
[11]	2	26 × 11 × 0.8	28, 39	>25	>5	>90	<10^−4^	-	-	-	-
[12]	2	32 × 32 × 1.59	2.475, 3.47	>15	5.9	81	<0.02	-	-	-	-
[13]	2	22 × 26 × 0.8	3.1–11.8	>20	3–6.6	>85	<0.03	-	-	-	-
[14]	2	18 × 21 × 0.8	2.8–12.2	>25	2–5	-	<0.013	-	-	-	-
[15]	2	27.65 × 12 × 0.203	28, 38	>22	>5.2	>91	<10^−5^	9.99	<−10	<0.4	-
[17]	2	60 × 100 × 0.813	28	>18	>5	>85	<0.0005	9.99	-	0.15	-
[18]	2	30 × 15 × 0.203	28, 38	>32	5.7	-	<10^−4^	10	-	<0.3	-
[19]	2	7.5 × 8.8 × 0.25	28, 38	>20	>5.3	-	<0.01	9.98	-	-	-
[20]	2	15.3 × 8.5 × 0.79	28, 38	>25	-	>83	-	-	-	-	-
[21]	4	158 × 77.8 × 0.381	28, 37, 38	>17	7.2	>70	-	-	-	-	-
[22]	4	104 × 104 × 0.51	2.45, 2.6, 5.2, 24, 28	>16	>6	>85	-	-	-	-	-
[23]	4	19.25 × 26 × 0.79	28, 38	>25	>5.72	>86	<0.004	-	-	-	-
[24]	4	50 × 12 × 0.8	25.1–37.5	>22	>6	>80	<0.001	-	-	-	-
[26]	4	30 × 35 × 0.76	25.5–29.6	>17	>6	>80	<0.01	9.96	-	<0.4	<−6
[27]	4	30 × 35 × 0.787	28	>40	>8	>80	<0.0003	9.96	-	<0.4	<−5
[28]	4	30 × 30 × 1.575	28	>30	>6	>90	<0.0005	9.99	-	<0.15	<−6
[29]	4	28 × 28 × 0.79	28, 38	>25	>4	-	<0.001	9.98	-	<0.5	<−5.5
[30]	4	20 × 20 × 0.79	28, 38	>24	>6	>83	<0.0001	9.99	-	<0.4	-
Pro.	4	12 × 11.6 × 0.508	27.1, 48.7	>27	>6	>90	<10^−6^	9.99	<−10	<0.03	<−3

## Data Availability

Not applicable.

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
