# Peer review of "Design and Analysis of Modified U-Shaped Four Element MIMO Antenna for Dual-Band 5G Millimeter Wave Applications"

_micromachines, 2023, doi:10.3390/mi14081545_

Round 1
Reviewer 1 Report
This paper considers 5G mmWave N257/N258 and N262 band applications and proposes a novel compact slotted four-element multiple input multiple output (MIMO) planar 20 monopole antenna. Experimental and simulation results show that the proposed antenna is well-suited for 5G mmWave communication applications. The paper is well written, the proposed antenna is original and may have applications in practice. However, the following points need to be addressed before the paper can be accepted for publication.
1.In introduction, the authors should briefly explain why the proposed antenna can achieve improved performance.
2. Why the two rectangular slots and the rectangular strip added to each radiating patch can improve the impedance performance of the antenna? How did the authors determine the sizes of the the rectangular slots and strip?
3. In Figure 4, what do the regions of different colors represent in the evolution process of the antenna?
4. Table 1 compares the proposed antenna with a few others on a number of performance indices. However, the table contains a large amount of data and does not show the overall performance of the proposed antenna clearly. It would be much better if all compared antennas can be ranked on each performance index and the average rank of the proposed approach is calculated and shown in the table.
5. In Section 3, a brief discussion on the strengths and weaknesses of the proposed antenna should be presented.
6. A brief discussion on potential future work should be presented in Conclusion.
Author Response
Dear reviewer,
Thank you for your comments. Please find the attached PDF file, which includes the answers to your concerns.
Best regards,

Reviewer 2 Report
1. In page 3 line100,generally the Rogers RT/duroid 5880 should have a thickness of 0.508 mm not width.
2. In page 11 line 224-225 and Page 12 line 234, please clearly indicate the simulated and measured gain at both 27.1 GHz and 48.7 GHz, respectively.
3. For evolution stages, from antenna B to antenna C, how about the S-parameter of the top layer of antenna C with full ground?
4. Generally, the soldering of SMA connectors and measurement cables would result in lots of loss especially at such high frequency over 20 GHz. So why the proposed antenna is still soldered? What kind of cables used for achieving measured radiation pattern? How long the cables are?
5. During measurement, the other two ports are connected to the load of 50 Ω when two ports were tested.
Please do a careful proofreading.
Author Response

(The authors gave the same response as above.)

Reviewer 3 Report
In this paper a U-Shaped 4-Element MIMO Antenna for Dual-Band 5G mmWave Applications is designed, fabricated and measured. This paper is well organized and can be considered for publications after some modifications listed as follows:
1- It is better “mmwave applications” in Title replaced with “millimeter wave applications”.
2- The reported size should be provided in both mm and lambda (λ).
3- Equations which are not belongs to authors should be cited.
4- In lines 56-59 mutual coupling reduction methods are investigated. Complete this section and add some related works maybe below paper is helpful:
“Mutual coupling reduction in microstrip patch antenna arrays using simple microstrip resonator. Wireless Personal Communications. 2022 Sep;126(2):1665-77.”
5- In Fig 10(a) there is significant difference between measured and simulated results (about 20 dB) in the second operating frequency , but in lines 207-28 mentioned that “there is a minor difference due to SMA connection losses, fabrication tolerances, dielectric, and conducting losses”. This 20 dB difference not belong to the mentioned reasons.
6- The quality of reported figures in Fig 10(c), (d) and 11(c) should be improved.
Overall, the presented paper is well written and it seems revised previously, where corrections are highlighted in the text.
​The device is fabricated and measured. The measured results validate the simulations. The paper can considered for publications after some minor modifications.
Minor editing of English language required
Author Response
Dear Reviewer,
Thank you for your comments. The paper has been revised carefully based on your comments. Please find the attached file, which contains the answers to your concerns.
Best regards,

Round 2
Reviewer 1 Report
The paper has been carefully revised to address all issues. I have no other concerns and recommend the acceptance of the paper.
Author Response
Thank you.
Reviewer 2 Report
The manuscript is improved. Generally, the measured gain is lower than the simulated gain due to some loss. From Figure 11 (a) and (b), the measured and simulated E-Field and H-field are not totally equal, so the simulated and measured gain should not be same. Please explain why both the measurement and simulation gains are 6.03 dBi at 27.1GHz and 5.95 dBi at 48.7 GHz.
The quality of English is generall fine.
Author Response

(The authors gave the same response as above.)

Round 3
Reviewer 2 Report
Thank authors for revising the manuscript. Figures can be smaller to make sub-figures shown in one page.
It would be better to do a careful proofreading before publishing.